# Effect of Flowering Time-Related Genes on Biomass, Harvest Index, and Grain Yield in CIMMYT Elite Spring Bread Wheat

**DOI:** 10.3390/biology10090855

**Published:** 2021-09-01

**Authors:** Susanne Dreisigacker, Juan Burgueño, Angela Pacheco, Gemma Molero, Sivakumar Sukumaran, Carolina Rivera-Amado, Matthew Reynolds, Simon Griffiths

**Affiliations:** 1International Maize and Wheat Improvement Center (CIMMYT), Texcoco 56237, Mexico; J.Burgueno@cgiar.org (J.B.); R.A.Pacheco@cgiar.org (A.P.); gemma.molero@kws.com (G.M.); S.Sukumaran@cgiar.org (S.S.); A.Rivera@cgiar.org (C.R.-A.); M.Reynolds@cgiar.org (M.R.); 2KWS Momont Recherche, 59246 Mons-en-Pévèle, Hauts-de-France, France; 3John Innes Center, Norwich NR4 7UH, UK; simon.griffiths@jic.ac.uk

**Keywords:** wheat, flowering time, gene-based markers, biomass, harvest index, grain yield

## Abstract

**Simple Summary:**

Allelic variants of vernalization (*Vrn*), photoperiod (*Ppd*), and earliness per se (*Eps*) genes in two panels of elite spring wheat were used to estimate their effects on the phenological stages, biomass (BM), harvest index (HI), and grain yield (YLD). Major spring alleles of *Vrn-1* had the largest effect on shortening the time to anthesis, while the Ppd-insensitive allele *Ppd-D1a* had the most significant positive effect on YLD. Furthermore, alleles at recently identified loci *TaTOE-B1* and *TaFT3-B1* promoted between 3.8% and 7.6% higher YLD and 4.2% and 10.2% higher HI in the two panels. Further, when the possible effects of the *TaTOE-B1* and *TaFT3-B1* alleles on the sink and source traits were explored, the favorable allele at *TaTOE-B1* showed positive effects on several sink traits related mainly to the grain number. Favorable alleles at *TaFT3-B1* followed a different pattern, with positive effects on the traits related to grain weight. The results of this study expanded the wheat breeders’ toolbox in the quest to breed better-adapted and higher-yielding wheat cultivars.

**Abstract:**

Grain yield (YLD) is a function of the total biomass (BM) and of partitioning the biomass by grains, i.e., the harvest index (HI). The most critical developmental stage for their determination is the flowering time, which mainly depends on the vernalization requirement (*Vrn*) and photoperiod sensitivity genes (*Ppd*) loci. Allelic variants at the *Vrn*, *Ppd*, and earliness per se (*Eps*) genes of elite spring wheat genotypes included in High Biomass Association Panel (HiBAP) I and II were used to estimate their effects on the phenological stages BM, HI, and YLD. Each panel was grown for two consecutive years in Northwest Mexico. Spring alleles at *Vrn-1* had the largest effect on shortening the time to anthesis, and the Ppd-insensitive allele *Ppd-D1a* had the most significant positive effect on YLD in both panels. In addition, alleles at *TaTOE-B1* and *TaFT3-B1* promoted between 3.8% and 7.6% higher YLD and 4.2% and 10.2% higher HI in HiBAP I and II, respectively. When the possible effects of the *TaTOE-B1* and *TaFT3-B1* alleles on the sink and source traits were explored, the favorable allele at *TaTOE-B1* showed positive effects on several sink traits mainly related to grain number. The favorable alleles at *TaFT3-B1* followed a different pattern, with positive effects on the traits related to grain weight. The results of this study expanded the wheat breeders’ toolbox in the quest to breed better-adapted and higher-yielding wheat cultivars.

## 1. Introduction

Within less than 10,000 years, wheat cultivation has expanded from its primary area of evolution within the Fertile Crescent to a broad spectrum of agroecologies around the globe, making it the most widely grown crop in the world [1]. Wheat is one of the cornerstone crops for global food security, offering approximately 20% of the calories and proteins of the human diet [2]. While the rate of world population growth is, in general, slowing down, it will still reach 8.6 billion people in 2030 and 9.8 billion by 2050 [3]. Thus, meeting the global demand for wheat will require a substantial increase in grain yield production per unit area. Moreover, new environments resulting from climate change will compromise wheat production [4], and further understanding of the genetic mechanisms related to the adaptive success of wheat is key for stable grain production in the future.

One essential path for wheat to achieve adaptation is through variations in its phenology, which is a critical component, as the grain yield performance is strongly influenced by the timing of the phenological stages in each particular environmental condition [5]. For instance, in order to reach the maximum seed size and number (potential yield), wheat must become established, develop a biomass, and flower at a time that coincides with the optimal seasonal conditions [6]. Phenology in wheat is mainly controlled by genes that regulate three pathways: the vernalization (Vrn), photoperiod (Ppd), and earliness per se (Eps) pathways. Vernalization is the acquisition of a plant’s ability to flower by exposure to cold [7]. According to the Vrn requirements, wheat is classified as having a winter or spring growth habit. Winter wheat has a considerable Vrn requirement, but spring wheat may be insensitive or only partly sensitive to Vrn [8]. The Ppd pathway promotes the floral transition in response to long days [9]. Ppd-insensitive wheat flowers independently of the day length and can be grown to maturity in long- or short-day environments. With the advances in molecular biology in previous decades, different alleles at the major *Vrn* and *Ppd* loci have been identified and shown to be responsible for affecting the flowering time and modifying the phenological stages [10,11,12]. The observed allelic variation has been associated with insertions, deletions, introductions of transposons, and other mutations in the promotor region and introns of *Vrn-1* and *Ppd-1* and, also, copy number variations (CNV) for the genes [13,14,15,16]. Alleles respond differently to environmental stimuli and act initially within separate pathways that converge at a point to produce flowers [17,18]. The third class of genes, which control the flowering time when both the Ppd and Vrn requirements are met, are the *Eps* genes [19]. They act in fine-tuning developmental patterns [20]; however, their genetics are still not well-understood. The underlying genes and causal polymorphisms have only recently been identified in hexaploid wheat. Zikhali et al. (2014 and 2016) [21,22] identified the *Eps-D1* locus in *Triticum aestivum* associated with advanced flowering. Furthermore, two additional QTL and their underlying candidate genes, *TaTOE-B1* and *TaFT3-B1*, related to the Ppd response under short days have been reported [23,24], which further broadens the allelic toolbox for manipulating and fine-tuning phenology in wheat.

Grain yield is a function of the total biomass (BM) and of partitioning the biomass by grains, i.e., the harvest index (HI). Several studies have indicated that genetic increases in grain yields have been driven mostly by increases in the biomass in modern wheat cultivars [25,26,27]. No systematic progress in the HI in wheat has been shown in recent decades [28], and the latest CIMMYT spring wheat releases expressed increased biomass, as well as grain yield, but decreased HI, precluding the full expression of yield potential [27]. In order to translate improvements in biomass production into gains in the yield potential, the HI must be maintained or ideally increased in high-biomass cultivars [29]. The areas of research related to both components, the BM and HI, are broad and include, for example, an increase of photosynthetic capacity, factors that influence the partitioning of assimilates, spike fertility, and, also, modifying crop phenology [28]. While crop phenology and grain yield are optimized with different variety × sowing date combinations and the varietal suitability in a particular growing environment, very little is still known about how the genes that synchronize the developmental phases within a growing environment affect the actual formation of a yield.

In this study, we aimed to (1) estimate the allele effects at the major and more recently identified *Vrn*, *Ppd*, and *Eps* genes on the phenology phases in two elite panels of CIMMYT elite spring bread wheat and (2) evaluate the possibility of maximizing the HI by controlling the duration of crop phenology by testing the allele effects of the same genes on the HI and several of its physiological components.

## 2. Materials and Methods

### 2.1. Plant Materials

High Biomass Association Mapping Panels (HiBAP) I and II were used for this study. Each panel consists of 150 elite spring bread wheat types, including high-yielding elite materials, pre-breeding lines crossed and selected for high yield and biomass, synthetic-derived and landrace-derived lines, and appropriate checks (Appendix A). Each panel was evaluated in two consecutive growing seasons at the Norman E. Borlaug Research Station (CENEB) in Ciudad Obregon, Sonora, Mexico. HiBAP I was evaluated during the winter cycle in the years 2015 to 2016 and 2016 to 2017 and HiBAP II during the 2017 to 2018 and 2018 to 2019 winter cycles. Fifty-seven lines in HiBAP I were in common with HiBAP II. HiBAP I included one durum line as a local check, which was excluded from the analyses.

### 2.2. Trait Measurements

Each HIBAP panel was evaluated for a series of agronomic and physiological traits using an alpha (0,1) experimental design under full irrigation [30,31]. The HIBAP I design for year 1 comprised 15 blocks of size 10, while, for the other trials (HIBAP I year 2 and HIBAP II both years), the experimental design had 30 blocks with 5 plots per block. All trials were sown in the autumn (from 24–27 November) for each winter cycle. The temperature was recorded by a meteorological station placed near the trials. The day length was estimated based on the latitude at CENEB (Appendix A). The experimental designs included four and three replications in HIBAP I and II, respectively. The lines were planted in raised beds (two beds per plot, each 0.8 m wide) with four (HiBAP I, 2015 to 2016) or two rows (HiBAP I 2016 to 2017, HiBAP II, 2 years) per bed (0.1 and 0.24 m between rows, respectively) 4 m long. Appropriate weed disease and pest controls were implemented to avoid yield limitations.

The phenological stages recorded were the initiation of booting (GS41, DTInB), anthesis (GS65, DTA), and physiological maturity (GS87, DTM), according to Zadok’s scale [32]. Four developmental phases were defined as follows: the vegetative growth period (VGP) from planting to DTA and grain-filling duration (GFD) from DTA to DTM; the VGP was further divided from planting to DTInB (DTInB) and from DTInB to DTA (DTInB-DTA). The percentage of grain filling (PGF) was calculated as GFD/DTM × 100. For each plot, the duration in days from emergence to these stages was calculated. The thermal time (TT) was computed by summing the average daily temperatures, following Angus et al. (1981) [33].

The biomass was measured between 40 and 42 days after emergence (BM_E40), at initiation of the booting stage, according to the plot phenology (BM_InB), approximately 7 days after anthesis (BM_A7) and after physiological maturity (BM_PM). In HIBAP I, all BM measures were taken in only two of the four replications. The samplings for BM_E40, BM_InB, and BM_A7 consisted of the total aboveground tissue 0.4 m^2^ from two beds starting at least 50 cm from the end of the plot (or the previous harvest) to avoid border effects. A subsample of the fresh biomass was weighted and oven-dried at 70 °C for 48 h for the constant dry weight measurement. At physiological maturity, a sample of 50 or 100 fertile shoots was taken randomly from the harvested area to estimate the HI. Grain yield (YLD) was determined in 3.2–4 m^2^ using the standard protocols [34]. BM_PM was calculated from YLD/HI.

Since HIBAP I and HIBAP II were phenotyped to perform genomic studies within the International Wheat Yield Partnership (IWYP), many additional agronomic and physiological traits were evaluated, including the sink- and source-related traits. As for the flowering time alleles found to promote the HI and YLD, their effect on several sink and source traits was additionally tested. The sink-related traits included and reported were the thousand-grain weight (TGW), number of grains per m^2^ (GM2), number of spikes per m^2^ seven days after anthesis (SpkM2_A7) and at physiological maturity (SM2), stems per m^2^ (StM2) 40 days after emergence (StM2_E40), initiation of booting (StM2_InB) seven days after anthesis (StM2_A7), number of grains per spike (GSP), grain weight per spike (GWSP) and spikelet (GWSPKL), grain filling rate (GFR), number of spikelets per spike (SPKLSP), number of infertile spikelets per spike (InfSPKLSP), spike lengths (Spk_L), fruiting efficiency (FE), and chaff dry weight of each individual spike (chaffDW_ind). The source-related traits included all the biomass samplings (BM_E40, BM_InB, BM_A7, and BM_PM); green area seven days after anthesis (GA_A7); lamina partitioning index (LamPI); lamina dry weight (LamDW); and crop growth rate pre- and post-heading (CGRpre and CGRpost), respectively. For some traits, the sink or source status of the referred organs can change depending on the growth stage and are therefore complex to classify. These traits included the internode 3 length (Int3_L), peduncle length (Ped_L), spike dry weight (SpkDW) seven days after anthesis (SpkDW_A7) and at physiological maturity (SpkDW_PM), spike dry weight per m^2^ seven days after anthesis (SpkDWM2_A7) and at physiological maturity (SpkDW_PM), stem dry weight (StDW) seven days after anthesis (StDW_A7) and at physiological maturity (StDW_PM), and StDW per m^2^ seven days after anthesis (StDWM2_A7) and at physiological maturity (StDWM2_PM). A more detailed description of how these traits were measured is given in Appendix A. Since the plant height was not significantly associated with the phenological stages and YLD among the panels, it was not considered as an additional trait.

### 2.3. Genotyping

The DNA isolation of both panels was performed from young leaf samples following a modified CTAB method [35]. A set of molecular markers associated with the *Vrn*, *Ppd*, and *Eps* alleles were applied on all the lines (Appendix A). The molecular markers were related to (i) the well-known alleles of major effect genes *Vrn-1*, *Vrn-3*, and *Ppd-1*; (ii) the genes recently identified by Zikhali et al. (2014, 2016, and 2017) [21,22,23], which effect sizes across the elite germplasm are still unknown; and (iii) the copy number variants (CNV) shown to be associated with the altered flowering time in several recent studies [14,36].

The Sequence-Tagged Sites (STS), Kompetitive Allele Specific PCR (KASP), and TaqMan^®^ CNV assays were used for genotyping. For the STS marker polymorphisms, the polymerase chain reaction (PCR) assay reaction mixtures in single 10-μL reactions used to amplify all the primers contained final concentrations of 1 × Buffer with Green Dye (Promega Corp., Madison, WI, USA), 200-μM deoxynucleotide triphosphates, 1.2-mM magnesium chloride, 0.25-μM of each primer, 1U of DNA polymerase (GoTaq^®^ Flexi, Promega Corp., Madison, WI, USA), and 50 ng of DNA template. The PCR profile was 94 °C for 2 min, followed by 30 cycles of 94 °C for 1 min, 54–60 °C for 2 min (dependent on the primer), and 72 °C for 2 min. The amplified products were separated on 1.2% agarose gels in tris-acetate/ethylene-diaminetetraacetic acid (TAE) buffer. The KASP markers were run in reactions containing 2.5-mL water, 2.5-mL PACE™ Genotyping Master Mix (https://3crbio.com/ (accessed on 23 January 2021)), 0.07-mL marker assay mix, and 50 ng of dried DNA with a PCR profile of 94 °C for 15 min of activation time, followed by 20 cycles of 94 °C for 10 s, 57 °C for 5 s, and 72 °C for 10 s and followed by 18 cycles of 94 °C for 10 s, 57 °C for 20 s, and 72 °C for 40 s. the fluorescence was read as an end point reading at 25 °C. The copy number variants (CNV) were applied only in HiBAP I and for the two genes, *Ppd-B1* and *TaFT3*, using TaqMan^®^ CNV assays as described in reference [14].

### 2.4. Summary Statistics

For both panels, the analysis of a standard lineal mixed model was conducted with the *lme* (R-project) procedure from META R 5.1 [37,38] with all the effects of years (Y), blocks within replications, replications within Y, replications, genotypes (G), and G × Y being considered as random effects. The broad-sense heritability (*H*^2^) was estimated in META R 5.1 considering all the terms in the model (Y, replications within years, G, and G × Y) as random effects. The broad-sense heritability (*H*^2^) was estimated for each trait over the two years, respectively, as:$$H^{2}=\frac{\sigma_{g}^{2}}{\sigma_{g}^{2}+\frac{\sigma_{ge}^{2}}{e}+\frac{\sigma_{}^{2}}{re}}$$
where *r* = number of replications, *e* = number of environments (Y), *σ*^2^ = error variance, $\sigma_{g}^{2}$ = genotypic variance, and $\sigma_{\mathrm{ge}}^{2}$ = G × Y variance. The phenotypic correlations (*r_p_*) between traits were simple Pearson correlations, and the genetic correlations between traits were calculated using the following formula:$$\rho_{g}=\frac{\bar{\sigma_{g\left(i,i^{\prime }\right)}}}{\bar{\sigma_{g\left(i^{}\right)}\sigma_{g\left(i^{\prime }\right)}}},$$
where $\bar{\sigma_{g\left(i,i^{\prime }\right)}}$ is the arithmetic mean of all pairwise genotypic covariances between traits *i* and *i*′ across the environments, and $\bar{\sigma_{g\left(i^{}\right)}\sigma_{g\left(i^{\prime }\right)}}$ is the arithmetic mean of all pairwise products of the variances among the traits across the environments.

### 2.5. Single Marker Regression and Stepwise Regression

To analyze the effect of the known genes in the different traits, simple linear regression was performed for each marker using the Best Linear Unbiased Predictors (BLUPs) of each trait across the two consecutive years in each panel. Due to the linkage between the genes and some degree of collinearity, a multivariate linear regression considering all gene simultaneously and using a stepwise selection procedure was performed to estimate each gene effect after removing the effects of the other genes. For stepwise selection, the ‘stepAIC’ function of MASS library in R was used.

## 3. Results

### 3.1. Phenotypic Variation of Trait Measurements

The ANOVA results of both HiBAP panels indicated significant variations among the G, environments (Y), and G × Y interactions for most traits, where the factor year was the least significant (Table 1). The full growing cycle of HiBAP II was, on average, eight days longer than the cycle of HiBAP I, using the BLUPs across the two consecutive years of each panel. The longer growing period in HiBAP II was reflected in both the developmental phases. The VGP/DTA and GFD in HiBAP I were, on average, 76.4 and 38.5 days long, while, in HiBAP II, they were 82.1 and 41.0 days long. The longer growing period also led, on average, to 353-kg/ha higher YLD in HiBAP II. The average daily temperatures from emergence to anthesis were slightly higher in HiBAP II than in HiBAP I, mainly due to higher maximum daily temperatures, while it was the opposite during GFD (Appendix A). The TT during the critical stage from the initiation of booting to anthesis (TTInB–TTA) was lower in HiBAP II (Table 1). The biomass at physiological maturity was similar in both panels, while the biomass measured in earlier stages (BM_E40, BM_InB, and BM_A7) were higher in HiBAP II than in HiBAP I. The harvest index in the two panels was also similar. The maximum HI measured in any of the two growing seasons was 0.52. The broad-sense heritability of the traits was, in general, medium to high. The highest heritabilities were estimated for HI, DTInB, and VGP/DTA in both panels. The heritability for YLD was 0.60 in HiBAP I and 0.83 in HiBAP II, respectively (Table 1).

The phenotypic and genetic correlations among the same traits are shown in Appendix A. In both datasets, BM_PM was the highest-correlated with YLD, followed by HI. A longer VGP/DTA was slightly negatively correlated with YLD and negatively correlated with HI.

### 3.2. Frequencies of the Vrn, Ppd, and Eps Alleles

The results of the molecular marker analysis that identified the alleles related to the flowering time genes are shown in Figure 1 and Appendix A. Overall, the frequency distribution of the alleles was similar in both panels (HiBAP I and II). Among the known major effect genes, the spring allele at *Vrn-B1* (*Vrn-B1a*) and *Vrn-D1* (*Vrn-D1a*) and the Ppd-insensitive allele at *Ppd-D1* (*Ppd-D1a*) were highly frequent (≥87%). Additionally, strong selection pressure against the spring allele at *Vrn-A1* (*Vrn-A1a*) was apparent in both panels. From the alternate winter alleles at *Vrn-A1*, allele *vrn-A1w*, characterized by an SNP leading to an amino acid change from leucine to a phenylalanine in exon 4 (previously identified in CIMMYT Veery lines (Eagles et al. 2011) [39], was more frequent (62% and 59% in HIBAP I and II, respectively) than the alternative allele *vrn-A1v*. At the *Ppd-A1* and *Ppd-B1* loci, the marker results indicated the presence of Ppd- sensitive alleles but a large variation of CNV at *Ppd-B1*.

Among the more recently identified flowering time genes (*TaFT3-B1*, *TaFT3-D1*, *TaTOE-B1*, and *Eps-D1*), the alleles that are designated to trigger earlier flowering were overall more frequent. The highest allele frequencies were observed for the Avalon type allele at *TaTOE-B1*, while the lowest frequency for the Spark type allele was at *TaFT3-B1*. At the *Eps-D1* locus, the lines showing a deletion at wheat chromosome 1DL (e.g., identified in the winter wheat lines Spark or Cadenza) were less frequent than the lines not carrying the deletion.

The assays to determine the CNVs were only deployed in HIBAP I. The assays resulted in a significantly higher allele variation than the related KASP assays, which are only bi-allelic. Eight and nine CNVs were observed for the *Ppd-B1* and *TaFT3-B1* TaqMan^®^ assays, respectively. While the CNVs were not correlated with the KASP marker results for *Ppd-B1*, they were correlated with the KASP for *TaFT3-B1*. The Avalon-type allele at *TaFT3-B1* was mainly identified in the lines with CN0 or CN1, while the Spark and Rialto-type alleles mainly carried larger CNVs.

### 3.3. Allele Effects on Phenological Phases

Single and stepwise marker regression identified the spring allele *Vrn-A1a* to have the largest significant effect on the phenological stages in HIBAP I and II, followed by the spring alleles *Vrn-D1a* and *Vrn-B1a* (Figure 2 and Appendix A). All three alleles most affected the earliest phenological phase assessed at the initiation of booting (DTInB). Only the presence of the *Vrn-A1a* allele led also to a significant longer GFD in both panels. The type of winter wheat allele at the *Vrn-A1* locus also significantly affected the phenological phases, while allele *vrn-A1v* shortened the VGP/DTA and elongated the GFD in HiBAP II but to a much lower extent than the spring allele. The *Ppd*-insensitive allele *Ppd-D1a*, the two-copy variant at *Ppd-B1*, and the Rialto type allele at *TaFT3-D1* elongated the time from DTInB to DTA in HiBAP I or II. The two latter alleles resulted in an elongated VGP/DTA, and only the two-copy variant at *Ppd-B1* led to a shorter GFD in HiBAP I.

### 3.4. Allele Effects on BM, HI and YLD

Single and stepwise marker regression were performed to evaluate the possibility that the flowering time alleles promote HI and/or YLD (Figure 3 and Figure 4 and Appendix A). The Ppd-insensitive allele *Ppd-D1a* had the most significant positive effect on YLD in HiBAP I and II, with a positive effect on BM_PM in HiBAP I and HI in HiBAP II, respectively. Significant positive effects on YLD were also detected by the spring alleles at *Vrn-D1* (*Vrn-D1a*) and *Vrn-B1* (*Vrn-B1a*) in HiBAP I but the latter to a much smaller extent. The *Vrn-D1a* allele showed a significant positive effect on BM_PM and no effect on the HI. Furthermore, the Avalon-type allele (which lengthens flowering under short days) at locus *TaTOE-B1* showed a significant positive effect on YLD in both germplasm panels. The alternative allele at the same locus (Cadenza-type, which shortens flowering under short days) showed the opposite negative effect on YLD in HiBAP I. The Avalon-type allele positively affected the BM measured at the initiation of booting (BM_InB) and the HI but negatively affected the BM measured at later stages (A7 and PM). The Cadenza-type allele, in addition to its negative effect on YLD, showed a negative effect on HI in HiBAP I.

Two alleles (Rialto and Avalon types) at the *TaFT3-B1* locus positively affected the YLD in HIBAP II. The two alleles showed a slightly negative effect on BM_A7 in both HiBAP and a positive effect on HI in HiBAP II. The third allele at the *TaFT3-B1* locus (Spark type) had a clear negative effect on the YLD in HiBAP II. The Timstein or Sonora-type allele (*Ppd-B1a (3x)*), an SNP related to a three-copy variant at *Ppd-B1* but not correlated with the CNV Taqman assay, showed a smaller positive effect on the YLD and HI in HIBAP II. The largest negative effects on the YLD were detected for the *Vrn-A1a* and *Vrn-B1b* alleles in HiBAP I, with a clear negative effect of the *Vrn-A1a* allele on BM_PM and no effect on the HI. Lastly, the *Eps-D1* Spark-type allele (temperature-sensitive, which shortens flowering) had a smaller negative effect on the YLD and BM.

To further verify the larger effects of the favorable alleles of the more recently identified genes *TaTOE-B1* and *TaFT3-B1* loci on the HI and YLD, a mean comparison of their respective alleles is shown in Table 2. The lines that carried the Avalon-type allele at *TaTOE-B1* yielded a 4.7–5.0% higher and showed a 3.3–8.0% higher HI than the lines carrying the Cadenza-type allele, while there was no significant difference in the means for BM_PM. The lines that carried the Avalon or Rialto-type alleles at *TaFT3-B1* (which lengthens and shortens flowering under short days) revealed a 1.2–6.0% higher YLD and a 3.1–9.6% higher HI than the Spark-type allele (which lengthens flowering under short days), while the means were only significantly different for the HI in HiBAP II. The combination of favorable alleles at both genes revealed means of 3.8–7.6% higher YLD and 4.2–10.2% higher HI than the combination of unfavorable alleles. The lines with one favorable and unfavorable allele each showed intermediate mean values indicating that both genes act additively.

### 3.5. Allele Effects of TaTOE-B1 and TaFT3-B1 on Source- and Sink-Related Grain Yield Traits

Due to their positive impact on both the HI and YLD, we further explored the possible effects of the alleles at the two genes, *TaTOE-B1* and *TaFT3-B1*, on the source- and sink-related grain yield traits. A schematic diagram, when the source- and sink-related traits considered in this study were produced, is given in Figure 5. The favorable Avalon-type allele at *TaTOE-B1* showed significant positive effects on several sink traits, mainly related to an increase in the grain number, i.e., GM2, FE, SPKLSP, StM2_E40, StM2_A7, and chaffDW_ind (Figure 6 and Appendix A). For the same and other sink traits, the Cadenza-type allele showed the opposite negative effects in HiBAP I, i.e., SM2, GSP, and InfSPKLSP. Furthermore, the GFR was increased, and the length of internode 3 (Int3_L) was reduced by the Avalon-type allele. In contrast, both *TaTOE-B1* alleles showed negative effects on the source traits LamDW_A7 and LamPI. The Avalon-type allele also showed an increased SpkDW_PM but a decreased StDW at different phenological stages (A7 and PM).

The allelic effects of the *TaFT3-B1* locus on the same source and sink traits followed a more complex pattern and were different from *TaTOE-B1* (Figure 7 and Appendix A). The favorable Avalon and Rialto-type alleles both showed significant positive effects on the sink traits, especially related to grain weight, i.e., TGW, GWSP, GWSPKL, and GFR. The unfavorable Spark-type allele had the opposite effect on the same traits in HiBAP II. The Avalon-type allele additionally favored the sink traits GSP and SPKLSP. In contrast to the *TaTOE-B1* gene, both the Avalon and Rialto-type alleles at *TaFT3-B1* showed negative effects on GM2, SM2, SkpM2_A7, StM2_E40, StM2_InB, StM2_A7, and chaffDW_ind. The Avalon-type allele additionally reduced InfSPKLSP. Like the sink traits, the impact of the *TaFT3-B1* alleles on the source traits was more complex and different in comparison to the *TaTOE-B1* alleles. The Avalon-type allele positively affected GA_A7, LamPI, and LamDW_A7, while the Rialto-type allele did not.

## 4. Discussion

Optimizing the phenological stages in a given growing environment is critical for wheat adaptation and the improvement of the yield potential. Although crop growth and development is a continuous succession of changes progressing towards maturity, the key developmental events around the phenological stages have been identified that mark important changes in the morphology and/or function of some organs [41,42]. The time at which the wheat crop reaches anthesis (DTA) is determined by the duration of the vegetative phase from sowing to the floral initiation (when leaves and tillers begin to appear), the early reproductive phase up to terminal spikelet initiation (when all spikelets are initiated and floret initiation starts), and the late reproductive phase of stem elongation ending with anthesis (when the number of florets is determined) [42]. All three phases are affected by three major environmental factors: Vrn, Ppd, and temperature *per se* [19]. The timely occurrence of anthesis therefore largely defines the yield potential of a genotype. There is a high degree of variability in the sensitivity to Vrn and Ppd among the genotypes, which is the likely reason for the wide adaptability of wheat to so many different environments. The Vrn n response varies from insensitive spring types, through facultative types, to extremely sensitive winter types [19]. Wheat is a long-day plant in which the rate of development is increased with longer day lengths. However, there are individual genotypes with varying sensitivities to Ppd, including insensitivity [43]. Additionally, a wealth of variations in Eps in different geographical regions has been shown [44]. Wheat breeding programs tailor the phenology of their germplasm for a particular environmental condition by testing different variety × sowing date combinations, thus combining specific sensitivities to the Vrn and Ppd with particular Eps characteristics across their genotypes. The increased knowledge of the genetic control of the determinants of phenological development could further facilitate breeding for improved yield potential in wheat, especially under the threats of global changes in the temperature and weather patterns that will continue to impede wheat production.

### 4.1. Estimations of the Allele Effects at Major and More Recently Identified Vrn, Ppd, and Eps Genes

The major *Vrn* and *Ppd* loci are usually quickly fixed in breeding programs targeting a specific selection environment. CENEB, near Ciudad Obregon, Mexico, is the key research and breeding site for the CIMMYT Global Wheat Program. Multi-year testing and controlled irrigation to create a simulation of diverse selection environments at CENEB have delivered novel high-yield potential germplasms worldwide [45], and research has shown that the conditions at this site represent major CIMMYT target regions for wheat breeding in the developing world [46,47,48].

Of the major effect genes, the two spring alleles, *Vrn-B1a* and *Vrn-D1a*, and the *Ppd-D1a*-insensitive allele were highly frequent and almost fixed in the two HiBAP panels and were also the most characteristic major gene combination in the annual globally distributed CIMMYT international nurseries (https://data.cimmyt.org/dataverse/root/?q=IBWSN (accessed on 30 July 2021)). Additionally apparent was a strong selection pressure against the spring allele *Vrn-A1a*. Among the *Vrn* genes, it is well-known that *Vrn-A1**a* has a strong effect on inhibiting the Vrn requirements and is epistatic to the dominant *Vrn-B1* and *Vrn-D1* genes [49,50,51]. In our study, *Vrn-A1a* significantly shortened the vegetative phase and early reproduction phase up to the initiation of booting (DTInB), which are the most sensitive phases for Vrn temperatures [42], resulting in a strong negative effect on the accumulation of BM and final YLD. The *Vrn-B1a* and *Vrn-D1a* alleles showed much smaller effects on the same developmental phases, resulting favorably for a YLD increase and suggesting that genotypes with some Vrn sensitivity are better adapted to CENEB. Stelmakh (1993 and 1998) [52,53] was one of the first who reported that the three major *Vrn* genes have a differential effect not only on flowering time but also on various yield components. In his studies, genotypes having two dominant alleles in combination at two *Vrn* loci tended to mature early with high yields, while the triple-dominant genotypes were found to be early maturing but low yielding. The incorporation of *Vrn-D1a* has generally been recommended in many spring wheat breeding programs. Van Beem et al. (2005) [54] already reported the predominance of the *Vrn-D1a* allele in CIMMYT elite spring bread wheat lines, and the Japanese wheat cultivar Akakomugi, thought to be the donor parent of this allele [55], was later transferred into early CIMMYT Green Revolution cultivars like Lerma Rojo 64 and Sonora 64. Additionally, the widely grown (8 to 9 million hectares globally) wheat cultivars Pastor, Attila, and Kauz possess *Vrn-D1a*. Our study thus confirms the results reported by Stelmakh (1993) [52] in the CIMMYT spring bread wheat germplasm but also extends its knowledge, as we were able to dissect the individual effects of major *Vrn-1* genes on the individual phenological stages BM, HI, and YLD. Of the two winter alleles at *Vrn-A1*, the *vrn-A1w* allele defined as the “Wichita”-type allele by Eagles et al. (2011) [39] was more frequent. The Wichita-type allele was suggested to confer a longer Vrn requirement than the alternate “Jagger”-type allele (*vrn-A1v*). In HiBAP II, *vrn-A1v* showed, as expected, a shorter Vrn requirement and, therefore, reduced the time to anthesis (VGP/DTA). The lines that carried the same allele in HiBAP I, however, showed a lower HI, which might explain the selected preference for *vrn-A1w*. The Ppd-insensitive allele *Ppd-D1a* in our study elongated the late reproductive phase or spike growth phase from DTInB to DTA in HiBAP I. The spike growth phase, taking place only a few days prior to anthesis, has been reported to be sensitive to Ppd responses [42]. The average daily temperatures during this phase were consistently increasing in HiBAP I. One could hypothesize that the lines with the *Ppd-D1a* allele might have reached this phase earlier at lower temperatures and, therefore, progressed more slowly compared to late-developing genotypes, for which this phase was occurring later and at higher temperatures. Slafer et al. (2001) [56] and Miralles and Slafer (2007) [57] hypothesized that a longer late-reproductive phase or spike growth phase increases the amount of biomass accumulated in spikes during stem elongation, and the final number of grains would be increased due to the reduced rates of floret death. The *Ppd-D1a* allele showed a positive effect on biomass accumulation (BM_PM) in HiBAP I and on HI in HIBAP II, resulting in a positive yield effect in both panels. The effect of *Ppd-D1a* that provides insensitivity to the Ppd during the pre-anthesis phase was also reported by Gonzalez et al. (2005) [58]) Two additional *Ppd* alleles elongated the late-reproductive phase, *Ppd-B1* (*CN2*) and *TaFT3-D1* (Rialto type). However, both alleles elongated this phase at the expense of earliness, resulting in a later time to anthesis and a shorter GFD, with no favorable effect on the YLD.

Recently identified candidate genes controlling the Ppd response in wheat were included in this study. Alleles at the two genes, *TaTOE-B1* (Avalon type) and *TaFT-B3* (Avalon and Rialto types), promoted the HI and YLD in both panels and in HIBAP II, respectively, which has not been tested and observed before. The favorable effects on the HI and/or YLD of these alleles was also confirmed in CIMMYT international screening nurseries (approx. 1300 lines) and in 197 F_4:6_-derived lines evaluated between 2017 and 2019 at CENEB (unpublished data). Both candidate genes were identified by Zikhali et al. (2017) [23] as controlling the short-day Ppd response. SNPs in exon 1 and 9 of the *TaTOE1-B1* gene were shown to separate earlier flowering UK cultivars Avalon, Rialto, and Charger from later flowering cultivars such as Spark or Claire. *TaTOE1-B1* was suggested to be a putative flowering time repressor, and the mutant allele (Avalon type) was expected to attribute earliness. In our study the mutant allele reduced the time to anthesis by 0.6 to 1 days in comparison to the wild-type allele in HiBAP I and II, respectively (data not shown); however, this effect was not significant.

The KASP marker related to *TaFT3-B1* was able to detect three alleles of the gene. The Rialto-type allele represents the intact form of the gene, while the Spark-type allele carries a point mutation and changes the highly conserved amino acid glycine (G) to serine (S) in the phosphatidylethanolamine-binding protein (PEBP) domain in exon 3. The gene is deleted in the UK cultivars Avalon or Charger [23]. *TaFT3-B1* is known to be a putative flowering promoter; Rialto was associated with early flowering, while the deletion of the whole gene, e.g., in Avalon, was associated with a later flowering phenotype. Single marker regression in HIBAP I supported the identified phenotypic variation from Zikhali et al. (2017) [23] and showed a moderate but significant effect of the Avalon-type allele on the DTA, while the wild-type allele of Rialto was associated with earlier flowering. The reported allele in Spark carrying the SNP that causes an amino acid change and is associated with later flowering was associated with later flowering in this study only in HIBAP II, with a clear significant negative effect on YLD.

### 4.2. Evaluation of the Possibility of Maximizing HI by Testing the Allele Effects on HI and Several of Its Physiological Components

Understanding the physiological changes responsible for the yield potential and the identification of the related prospective traits useful for future breeding is an active area of research and one of the main objectives of IWYP (https://iwyp.org/ (accessed on 30 July 2021)). Critical traits to be considered to further increase the yield potential must be related to increases in the sink size during grain filling, either by increasing the potential size of the grains or by further increasing the GM2. A large body of evidence has clearly established that GM2, although generated during the whole period from sowing to immediately after anthesis [19], is extremely responsive to changes in growth/partitioning during only a few weeks before anthesis, also referred to as the rapid spike growth period (RSGP) [57,59]. Other studies have shown that lengthening of the RSGP leads to increases in SpkDW at anthesis, resulting in genetic gains in the YLD [60,61,62,63].

With the focus on performing genetic studies for the agronomic and physiological traits related to the yield potential [30,31], both HiBAP panels were phenotyped for a series of sink- and source-related traits. We estimated the effect of the two Ppd response genes *TaTOE-B1* and *TaFT3-B1* on several of these traits. For the favorable allele at *TaTOE-B1*, a significant positive effect on GM2 was estimated, together with a larger number of StM2 (StM2_E40 and A7), more SPKLSP, enhanced FE, increased GFR, and greater SpkDW_PM and chaffDW_ind. We did not observe a significant increase in SpkDW at anthesis (SpkDW_A7) but a significant decrease in StDW_A7 and StDW_PM, which indicated that, in the presence of the favorable allele, more assimilates were diverted to the growing spikes, turning into an increase of GM2. ChaffDW at maturity (chaffDW_ind) is also sometimes used to estimate SpkDW_A7, while described to be consistently higher [64]. This was reflected in our study by a significant increase of chaffDW_ind in the lines carrying the Avalon-type allele at *TaTOE-B1* but a nonsignificant increase of SpkDW_A7. Various previous studies have reported genetic variations for FE and associations with GM2 [27,65,66,67]. The fruiting efficiency is the outcome of floret development. Slafer et al. (2015) [64] described two alternative physiological pathways to improve the FE: (i) an increased allocation of assimilates for the developing florets before anthesis, or (ii) a reduced demand of the florets for maintaining their normal development. Based on our results, we speculated that the Avalon-type allele at *TaTOE-B1* promoted the first pathway. Furthermore, the Avalon-type allele at *TaTOE-B1* desirably reduced Int3_L. Recently, Rivera-Amado et al. (2019) [68] revealed a positive effect of reduced internode length (internodes 2 and 3) on SpkDW at anthesis, which was positively associated with GM2 across 26 CIMMYT elite lines. The critical RSGP and percent grain filling (PGF) ratios were calculated in HIBAP I and II. The alleles at *TaTOE-B1*, however, did not show any significant effects on the RSGP, as would be expected from the physiological response (data not shown), while the Avalon-type allele showed a slightly longer percent grain filling (Appendix A). Thus, it still remains somewhat unclear how the flowering time alleles at *TaTOE-B1* cause the observed physiological changes, and more detailed studies of their effects on the individual developmental phases up to anthesis might be required.

The contrasting effect of the two genes *TaTOE-B1* and *TaFT3-B1* on the phenological stages (the first being a flowering repressor and the latter a promoter) was also reflected by the regression results on the sink-related traits. While the favorable allele at *TaTOE-B1* promoted GM2, the Avalon and Rialto-type alleles at *TaFT3-B1* boosted traits related to the grain weight, i.e., TGW, GWSP, GWSPKL, and GFR, in addition to Spk_L, GSP, and SPKLSP. In previous studies, variations in the grain weight were mainly related to GFR but, also, to GFD [69,70,71]. In this study, the favorable allele at *TaFT3-B1* had no effect on the GFD or PFG; however, the unfavorable Spark-type allele flowered later in HIBAP I, with significant negative effects on the YLD, GFR, and traits related to grain weight.

## 5. Conclusions

Our study improved our understanding of the relationship between the flowering time and HI in elite spring bread wheat lines. We identified that the two new candidate genes *TaTOE-B1* and *TaFT3-B1*, both controlling the Ppd response, promoted the HI in CIMMYT spring bread wheat grown at the main research station CENEB. Both genes act differently through complex physiological pathways. The gene effects should be further validated, and the development and evaluation of gene-specific near-isogenic lines in diverse CIMMYT elite backgrounds is underway. The gene effects should also be tested across a more diverse set of environments, in addition to CENEB. Our study expanded the wheat breeder’s toolbox in the quest to breed better-adapted and more resilient wheat cultivars. Recently, Hu et al. (2021) [72] (see Appendix A) used a gene-based phenology model to identify the optimal flowering time periods and optimal sowing dates for sites in irrigated environments, considering the impacts of frost and heat stress on the YLD. The gene-based model could predict the wheat phenology by the combination of a conventional process-based model and genetic information, which can consistently be updated with new relevant candidate genes such as *TaTOE-B1* and *TaFT3-B1* becoming available. An increased understanding of the optimal flowering period for a specific wheat production region could help breeders to select adaptable genotypes and guide farmers in modifying the management practices (sowing date and cultivar) to fit flowering and maximize the YLD.

## Figures and Tables

**Figure 1 biology-10-00855-f001:**
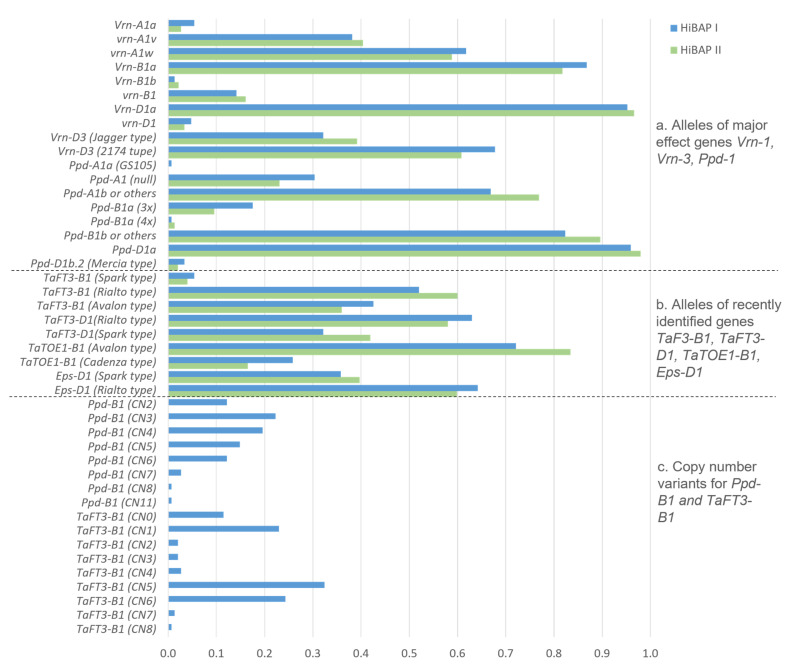
Allele frequencies of (**a**) the known major effect genes (Vrn-1, Vrn-3, and Ppd-1); (**b**) newly identified phenology genes; and (**c**) copy number variants in the High Biomass Association Mapping Panels (HIBAP) I and II.

**Figure 2 biology-10-00855-f002:**
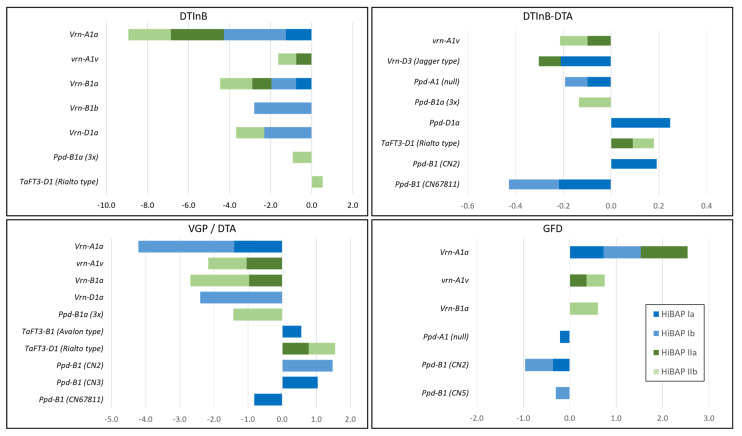
Significant effects (*p* 0.05) derived by single marker regression (HiBAP **I** and **IIa**) and stepwise marker regression (HiBAP **I** and **IIb**) for the alleles of the flowering time genes on the phenological phases in High Biomass Association Mapping Panels (HIBAP) I and II. DTInB: Days to initiation of booting, DTA: Days to anthesis, VGP: Vegetative growth period, and GFD: Grain filling duration.

**Figure 3 biology-10-00855-f003:**
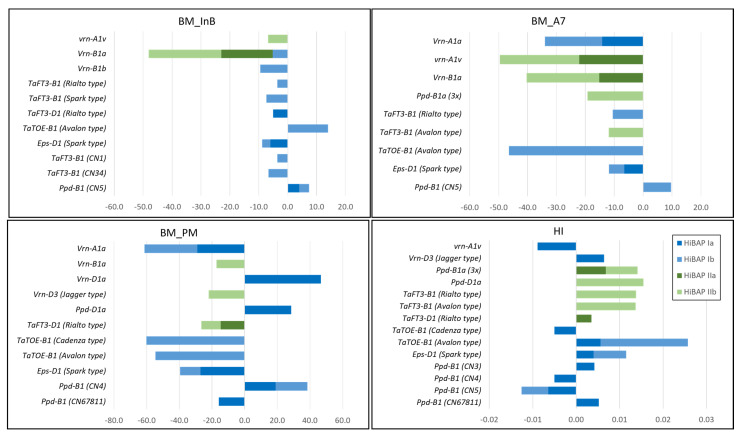
Significant effects (*p* 0.05) derived by single marker regression (HiBAP **Ia** and **IIa**) and stepwise marker regression (HiBAP **Ib** and **IIb**) for the alleles of the flowering time genes on the Biomass (BM) and Harvest Index (HI) in the High Biomass Association Mapping Panels (HIBAP) I and II. InB: Initiation of booting, A7: Seven days after anthesis, and PM: Physiological maturity.

**Figure 4 biology-10-00855-f004:**
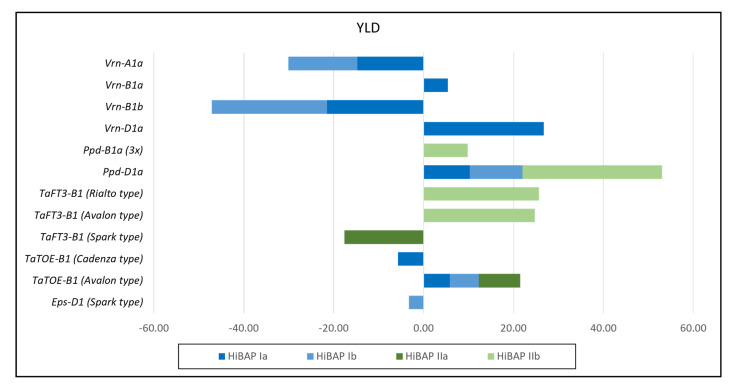
Significant effects (*p* 0.05) derived by single marker regression (HiBAP **Ia** and **IIa**) and stepwise marker regression (HiBAP **Ib** and **IIb**) for the alleles of the flowering time genes on the grain yield (YLD) in the High Biomass Association Mapping Panels (HIBAP) I and II.

**Figure 5 biology-10-00855-f005:**
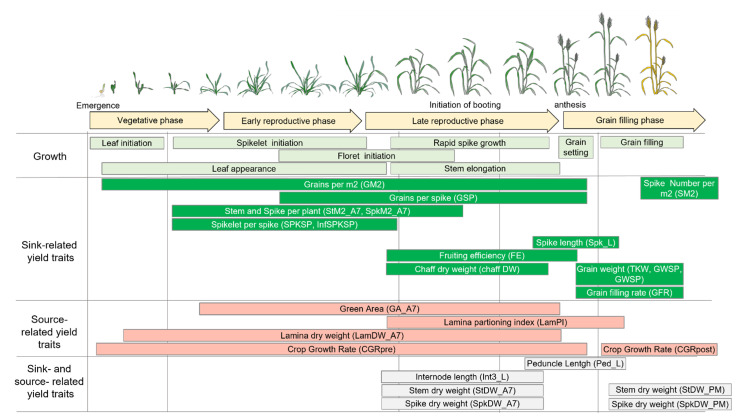
Schematic diagram of wheat growth [40] and when different source- and sink-related yield traits are produced. A full description of the traits is given in Appendix A.

**Figure 6 biology-10-00855-f006:**
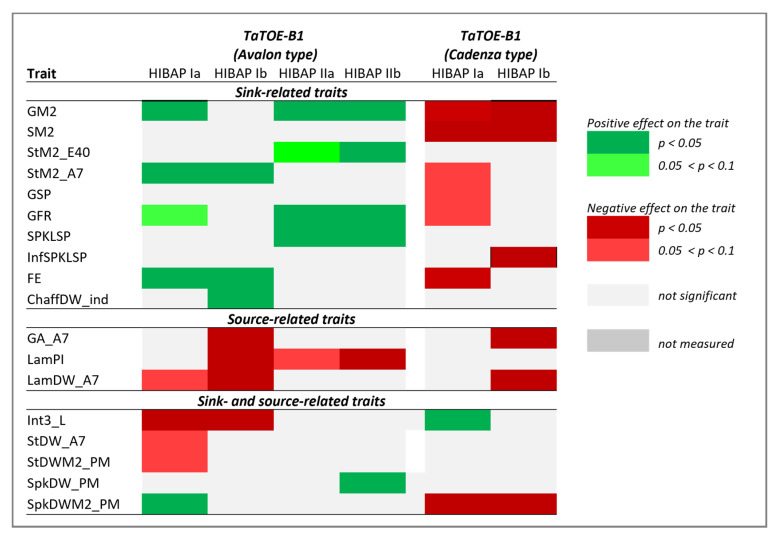
Significant effects derived from the alleles at *TaTOE-B1* estimated by single marker regression (HiBAP **Ia** and **IIa**) and stepwise marker regression (HiBAP **Ib** and **IIb**) for the alleles of the flowering time genes on the source- and sink-related grain yield traits in High Biomass Association Mapping Panels (HIBAP) I and II.

**Figure 7 biology-10-00855-f007:**
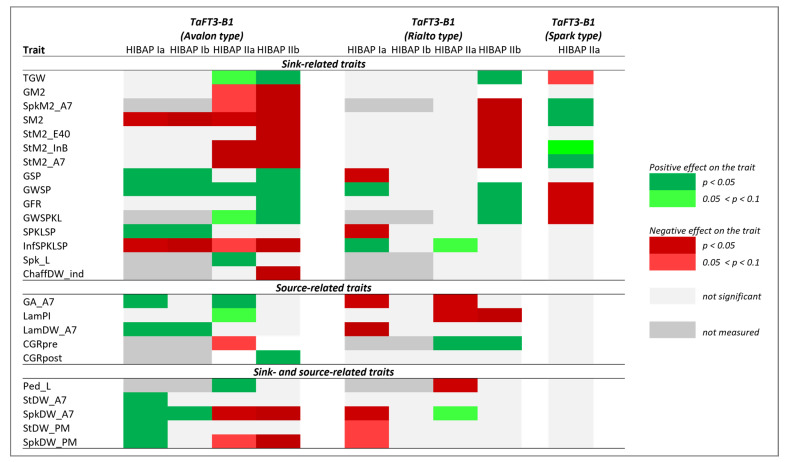
Significant effects derived from the alleles at *TaFT3-B1* estimated by single marker regression (HiBAP **Ia** and **IIa**) and stepwise marker regression (HiBAP **Ib** and **IIb**) for the alleles of the flowering time genes on the source- and sink-related grain yield traits in High Biomass Association Mapping Panels (HIBAP) I and II.

**Table 1 biology-10-00855-t001:** Descriptive statistics for the phenological stages BM, HI, and YLD in the High Biomass Association Mapping Panel (HiBAP) I and II, each grown for two years at CENEB, NW Mexico.

	HiBAP I						HiBAP II					
Trait	Mean	Range	H^2^	G	G × Y	Y	Mean	Range	H^2^	G	G × Y	Y
Developmental phases												
DTInB (days)	60.8	53.7–66.8	0.83	***	***	***	65.6	57.5–72.5	0.92	***	***	***
DTInB-DTA (days)	15.7	14.3–17.2	0.57	***	***	***	16.5	14.8–17.6	0.34	**	***	***
VGP/DTA (days)	76.4	69.4–84.1	0.87	***	***	***	82.1	73.7–90.6	0.93	***	***	***
GFD (days)	38.5	34.4–45.3	0.63	***	***	**	41.0	36.5–45.5	0.75	***	***	ns
TTInB	1007.4	886.0–1134.6	0.92	***	***	***	1142.5	995.8–1248.2	0.91	***	***	***
TTInB-TTA	279.9	248.8–319.7	0.55	***	***	***	255.6 ^1^	171.5–306.3 ^1^	-	-	-	-
TTVGP/TTA	1287.3	1142.9–1443.1	0.94	***	***	***	1398.1	1264.4–1535.6	0.91	***	***	*
TTGFD	716.6	678.0–796.5	0.64	***	***	***	742.7	689.4–794.0	0.58	***	***	***
Agronomic and physiological traits												
BM_E40 (g/m^2^)	146.6	136.0–156.9	0.32	ns	ns	ns	188.3	171.6–203.0	0.39	***	ns	***
BM_InB (g/m^2^)	424.9	387.2–453.7	0.24	**	***	*	569.4	466.6–652.6	0.61	***	***	ns
BM_A7 (g/m^2^)	857.7	778.9–945.7	0.62	***	ns	ns	1036.4	870.6–1185.3	0.70	***	***	ns
BM_PM (g/m^2^)	1355.1	1254.5–1472.4	0.41	**	***	*	1341.4	1125.1–1471.9	0.73	***	**	***
HI	0.47	0.42–0.51	0.73	***	*	ns	0.47	0.41–0.52	0.83	***	***	**
YLD (kg/ha)	5957	5303–6601	0.60	***	***	ns	6310	5111–7009	0.83	***	***	ns

DTInB: days to initiation of booting, DTA: days to anthesis, VGP: vegetative growth period, GFD: grain filling duration, TTInB: thermal time to initiation of booting, TTA: thermal time to anthesis, TTGFD: thermal time during grain filling duration, BM_E40: Biomass measured 40 days after emergence, BM_InB: Biomass at initiation of booting, BM_A7: Biomass measured 7 days after anthesis, BM_PM: Biomass at physiological maturity, HI: Harvest Index, YLD: Grain yield, H2: Broad-sense heritability, G: Genotypes, and Y: Years. ^1^ Best linear unbiased estimator (BLUEs) instead of the best linear unbiased predictor (BLUP), which failed for this trait. * *p* 0.05, ** *p* 0.01, *** *p* 0.001 and not significant (ns).

**Table 2 biology-10-00855-t002:** Means of the biomass measured at physiological maturity (BM_PM), the harvest index (HI), and grain yield (YLD) for the alleles identified with the molecular markers associated with the genes *TaTOE-B1* and *TaFT3-B1* and the allele combinations of both genes.

	*TaTOE-B1*		*TaFT3-B1*			*TaTOE-B1* + *TaFT3-B1*		
Panel	Avalon Type	Cadenza Type	Avalon Type	Rialto Type	Spark Type	Avalon + Avalon/Rialto	Cadenza + Avalon/Rialto	Cadenza + Spark Type
Allele frequency								
HIBAP I	0.72	0.26	0.43	0.52	0.05	0.73	0.23	0.04
HIBAP II	0.83	0.17	0.36	0.60	0.04	0.83	0.14	0.03
BM_PM								
HIBAP I	1349 a	1335 a	1355 a	1341 a	1378 a	1352 a	1328 a	1384 a
HIBAP II	1342 a	1327 a	1337 a	1346 a	1302 a	1344 a	1332 a	1302 a
HI								
HIBAP I	0.474 a	0.459 b	0.473 a	0.469 a	0.455 a	0.474 a	0.459 b	0.455 ab
HIBAP II	0.486 a	0.450 b	0.493 a	0.476 bc	0.450 bc	0.486 a	0.455 b	0.44 b
YLD (kg/ha)								
HIBAP I	6010 a	5740 b	6010 a	5910 a	5840 a	6020 a	5740 b	5800 ab
HIBAP II	6350 a	6050 b	6350 a	6310 a	5990 a	6360 a	6110 b	5910 b

Different letters after each mean indicate differences according to the LSD test at *p* 0.05.

## Data Availability

The genotyping and phenotyping data will be made available at the following repository: https://data.cimmyt.org/dataverse/iwypdvn (accessed on 30 July 2021).

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
