# Peer review of "Effect of Flowering Time-Related Genes on Biomass, Harvest Index, and Grain Yield in CIMMYT Elite Spring Bread Wheat"

_biology, 2021, doi:10.3390/biology10090855_

Round 1
Reviewer 1 Report
Very important research that combine the targeted to most significant characteristics e.g how allelic variants at verbalization, photoperiod, and Earliness effect biomass (BM), harvest index (HI), and grain yield (YLD). Very targeted research with huge practical results in applied wheat breeding, especially now that earliness is more important due to climate change.
It is a very very good manuscript that deserves to published.
Just two questions:
- Could please authors explain why they chose (Line 126, “two beds per plot each 0.8 m wide”) 0.8m wide
- How authors explain such high H2=0.83 about YLD in table 1
A grammar and English language control will help to the correction of a few errors in the manuscript.

Author Response
Many thanks for the positive response by the reviewer, below the answer to the two questions raised. We also send the manuscript for english editing.
1. Could please authors explain why they chose (Line 126, “two beds per plot each 0.8 m wide”) 0.8m wide
The raised bed planting system is routinely used at the CENEB station in Obregon and has been focus of CIMMYT agronomy research since many years (see also Sayre et al, 2005). The bed planting system has been adopted for several crops in the Yaqui Valley, around Obregon and in many other areas with similar cropping systems. Some advantages of the raised bed planting are reduced seed rates, less irrigation water, less lodging, better fertilizer management, options for mechanical weeding etc,..
CIMMYT yield trails before global dissemination though the International Wheat Improvement Network are planted in bed but also on flat for full comparison. However, other trials including HIBAP are planted in beds only.
Sayre KD, Limon-Ortega A, Govaerts B (2005) Experiences with permanent bed planting systems CIMMYT/Mexico. In: Roth CH, Fischer RA, Meisner CA (eds) Evaluation and performance of permanent raised bed cropping systems in Asia, Australia and Mexico. Proceedings of a workshop held in Griffith, Australia. ACIAR Proceedings 121, pp 12–25
2. How authors explain such high H2=0.83 about YLD in table 1
The CENEB station, located in the Sonora desert, is an ideal site for testing YLD potential. It is predictive of performance for most of the high yielding spring wheat agroecosystems below 45° latitude. Wheat is grown under irrigated conditions and without major disease problems except some leaf and stem rust. The environmental conditions at CENEB are considered optimum for wheat and provide the opportunity for maximum expression of yield. Usually, trials are very well managed. High heritabilities for YLD at CENEB are therefore not necessarily uncommon, see e.g., also Juliana et al 2020 or Mondal et al., 2020 which report heritabilities for YLD of 0.79 and 0.81, respectively.
Juliana, P.; Singh, R.P.; Braun, H.-J.; Huerta-Espino, J.; Crespo-Herrera, L.; Payne, T.; Poland, J.; Shrestha, S.; Kumar, U.; Joshi, A.K.; et al. Retrospective Quantitative Genetic Analysis and Genomic Prediction of Global Wheat Yields. Front. Plant. Sci. 2020, 11.
Mondal, S., Dutta, S., Crespo-Herrera, L., Huerta-Espino, J., Braun, H. J., and Singh, R. P. (2020), “Fifty years of semi-dwarf spring wheat breeding at CIMMYT: Grain yield progress in optimum, drought and heat stress environments,” Field Crops Research, 250, 107757
Reviewer 2 Report
This paper is taken a part to the increased knowledge on the genetic control of the determinants of phenological development and could further facilitate breeding for improved yield potential in wheat, especially under the threats of global changes in temperature and weather patterns that will continue to impede wheat production. An example, by this study they were able to dissect the individual effects of mayor Vrn-1 genes to the individual phenological stages, biomass, harvest index, and yield. Further, two additional Ppd alleles elongated the late reproductive phase, Ppd-B1 (CN2) and TaFT3-D1 (Rialto type). However, both alleles elongated this phase at the expense of earliness, resulting in a later time to anthesis and a shorter grain filling duration, with no favourable effect on yield. Recently identified candidate genes controlling photoperiod response in wheat were included in this study. Alleles at the two genes, TaTOE-B1 (Avalon type) and TaFT-B3 (Avalon and Rialto type), promoted har4vest index and yield in both panels and in HIBAP II, respectively, what has not been tested and observed before. Likewise, the effect of the two photoperiod response genes TaTOE-B1 and TaFT3-B1 on several physiological traits were examined. In conclusion, this study improved the understanding of the relationship between flowering time and harvest index in elite spring bread wheat lines. The authors identified the two new candidate genes, TaTOE-B1 and TaFT3-B1, both controlling photoperiod response promoted harvest index in CIMMYT spring bread wheat. The gene-effects should be further validated and the development and evaluation of gene-specific near-isogenic lines in diverse CIMMYT elite backgrounds is underway. Increasing understanding of the optimal flowering period for a specific wheat production region could help breeders to select adaptable genotypes, and guide farmers to modify management practices (sowing date and cultivar) to fit flowering and maximize yield.
Excellent work with high scientific value!
(newly added after the report is submitted:)
The group which submitted this manuscript is well known and present in the scientific area related to this topic; also the reference, Slafer e.g.. The requirements for understanding this kind of research is quite high. They create and handle questions, but were also able to give answers at high level.
Reseach and experiments related to the relationship between phenology of wheat(barley) and the three genes (Vernalization (Vrn), Photoperiod (Pdp) and earliness per se (Eps), (line 61/62) are scarce. [It is also a part in my lecture "Yield physiology"; after acceptance by the journal, I will introduce some of the new results for the students?.]
For a moment I was thinking that it could be necassary to explan (in M&M) the 'Avalon', 'Cadenza,'Rialto', 'Sparke' type; but later in the part 'Results' the explanations appears. All in one, from my side I have no comment or remark to improve this manuscript, which is 'smooth' from the beginning to the end.
Author Response
Thanks for the positive feedback on our manuscript. We did send the manuscript additionally for english editing to improve language.